# Surface enrichment and diffusion enabling gradient-doping and coating of Ni-rich cathode toward Li-ion batteries

Haifeng Yu[1,4], Yueqiang Cao [2,4], Long Chen[1], Yanjie Hu[1], Xuezhi Duan[2], Sheng Dai[3], Chunzhong Li [1] & Hao Jiang [1✉]

Critical barriers to layered Ni-rich cathode commercialisation include their rapid capacity fading and thermal runaway from crystal disintegration and their interfacial instability. Structure combines surface modification is the ultimate choice to overcome these. Here, a synchronous gradient Al-doped and $LiAlO_2$-coated $LiNi_{0.9}Co_{0.1}O_2$ cathode is designed and prepared by using an oxalate-assisted deposition and subsequent thermally driven diffusion method. Theoretical calculations, in situ X-ray diffraction results and finite-element simulation verify that $Al^{3+}$ moves to the tetrahedral interstices prior to $Ni^{2+}$ that eliminates the Li/Ni disorder and internal structure stress. The $Li^+$-conductive $LiAlO_2$ skin prevents electrolyte penetration of the boundaries and reduces side reactions. These help the Ni-rich cathode maintain a 97.4% cycle performance after 100 cycles, and a rapid charging ability of 127.7 $mAh\,g^{-1}$ at 20 C. A 3.5-Ah pouch cell with the cathode and graphite anode showed more than a 500-long cycle life with only a 5.6% capacity loss.

[1] Key Laboratory for Ultrafine Materials of Ministry of Education, Shanghai Engineering Research Center of Hierarchical Nanomaterials, School of Materials Science and Engineering, East China University of Science and Technology, Shanghai, China. [2] State Key Laboratory of Chemical Engineering, School of Chemical Engineering, East China University of Science and Technology, Shanghai, China. [3] Key Laboratory for Advanced Materials and Feringa Nobel Prize Scientist Joint Research Center, School of Chemistry and Molecular Engineering, East China University of Science and Technology, Shanghai, China. [4] These authors contributed equally: Haifeng Yu, Yueqiang Cao. ✉email: jianghao@ecust.edu.cn

The COVID-19 pandemic has promoted the development of Li-ion batteries (LIBs) globally. For example, an increased number of people required electronic products to work from home and attend remote conferences, and this phenomenon may be universalised in the post-pandemic era[1]. However, LIBs still experience a "low-energy anxiety"[2,3]. Layered Ni-rich oxides have been considered promising high-energy and power-dense cathode materials owing to their large theoretical capacity (270 mAh g$^{-1}$), high output voltage (3.7 V) and rapid ion/electron transfer[4–6]. To boost the packing density and reduce side reactions with electrolytes, Ni-rich cathode materials may be fabricated as spherical micron-sized secondary particles with nanosized primary particles[7]. However, phase transition from H2 to H3 with anisotropic volume deformation yields intergranular and intragranular microcracks, and continuous capacity attenuation of this material[8,9]. Local heat accumulation under high-rate operation accelerates the transformation from a layered structure to electrochemically inert rock salt phase, which affects the reaction kinetics and capacity deleteriously[10].

The cathode–electrolyte interface and the crystal structure must be stabilised simultaneously to obtain a high-power and stable Ni-rich cathode materials. Previous work introduced a surface coating to stabilise the Ni-rich material interface. However, their performance are still far from success because (1) the low Li ionic conductivity coating decreases the specific capacity;[11,12] and (2) the non-uniform coating does not protect the materials well[13,14]. Heteroatom doping was applied to stabilise the crystal structure of primary particles with an enhanced Li$^+$ diffusivity[15,16]. Because structure disintegration usually commences around the crystal surface[17,18], and considering the specific capacity, an effective surface-enrichment gradient doping is required for this cathode. Theoretical calculations revealed that the doping efficiency is generally low because of easily formed dopant-containing electrochemical inert compounds on the particle surface[19,20]. Because of the sensitivity of the Ni-rich oxide precursor to moisture and air, and abundant Li source (i.e., $Li_2CO_3$, LiOH)[21,22], the traditional multistep coating and doping approach is undesirable for industrial production. Up to date, there is no Ni-rich cathode materials can realise a good power and long-term behaviour to satisfy the industry requirements.

In this work, we demonstrate an ultrafast and highly stable performance of synchronous gradient Al-doped and $LiAlO_2$-coated $LiNi_{0.9}Co_{0.1}O_2$ (NCAl-LAO) cathode materials, which is achieved by an oxalate-assisted deposition method. Compared to previously reported single doping or coating modification[9–16,23], the simultaneously obtained gradient Al doping inside the primary particles and uniform $LiAlO_2$ coating on the surface of the secondary particles can concurrently stabilise crystal structure and hinder the parasitic reaction at the interface. This strategy is revealed to minimise the capacity sacrifice due to the incorporation of electrochemical inert element. This work addresses two key issues of crystal disintegration and interfacial instability of the Ni-rich cathode, and provides a dual-modification approach for high-energy cathodes.

## Results

**Failure mechanism and modification design of Ni-rich cathode.** $Ni^{2+}$ in Ni-based layered oxides tends to migrate to vacancies in the Li layer along the tetrahedral interstice after $Li^+$ extraction, accompanied by a loss of lattice oxygen (Fig. 1a)[24–26]. These parasitic reactions during charging promote unfavourable phase transformations and reduce the thermal stability. $Ni^{2+}$ oxidation at the Li layer causes a significant shrinkage of the octahedral cell because of the corresponding electron loss on the $e_g$ orbit (Fig. 1b). The unit cell shrinkage is associated closely with $c$-axis shrinkage, which causes anisotropic volume deformation of the primary particles with a

concomitant internal mechanical stress. Al element has been applied extensively to enhance the structural and thermal stabilities of Ni-rich cathodes in practical applications mainly because of its suitable diameter and high Al–O bond energy[27,28]. To verify the effect of $Al^{3+}$ doping, density functional theory (DFT) calculations were performed to disclose the migration and occupancy behaviour of $Al^{3+}$ and $Ni^{2+}$ in Al-doped Ni-rich cathodes. $Al^{3+}$ locates at the 3b site in transition metal (TM) layers (Supplementary Fig. 1 and Supplementary Table 1). As shown in Fig. 1c and Supplementary Table 2, the energy barrier to $Al^{3+}$ migration (0.19 eV/f.u.) from octahedral interstices ($O_{TM}$) sites to tetrahedral interstices in the Li layer ($T_{Li}$) sites is lower than that of $Ni^{2+}$ (0.33 eV/f.u.). Therefore, $Al^{3+}$ shows a higher energetic tendency to migrate from $O_{TM}$ sites to $T_{Li}$ sites during charging. The $Al^{3+}$ formation energy (–1.06 eV/f.u.) in the $T_{Li}$ sites is lower than that of the $Ni^{2+}$ (–0.70 eV/f.u.) (Fig. 1d and Supplementary Table 3), which indicates that $Al^{3+}$ is more inclined to remain in $T_{Li}$ sites instead of migrating to octahedral interstices in Li layer ($O_{Li}$) sites compared with $Ni^{2+}$. Therefore, $Al^{3+}$ will migrate preferentially from $O_{TM}$ and be trapped in $T_{Li}$ sites, which hinders the $Ni^{2+}$ migration path and reduces the $Ni^{2+}$ content at the Li layer. Besides the doping of $Al^{3+}$ in the crystal structure of Ni-rich cathodes, a lithium aluminium oxide ($LiAlO_2$) coating with $Li^+$ surface conductivity can stabilise the cathode–electrolyte interface in organic electrolyte (Fig. 1e), due to the high chemical stability of $LiAlO_2$ (Supplementary Fig. 2)[29,30]. Considering that the structural deterioration starts from the surface region, it is the optimal design to simultaneously achieve uniform $LiAlO_2$ coating and high-efficiency gradient Al doping for Ni-rich cathodes by a simple and scalable approach.

To this end, it is pivotal to achieve controllable and uniform surface deposition of Al-containing compounds on the surface of the precursors of Ni-rich cathodes, but this is also very difficult since aluminium compounds are prone to self-nucleation reactions in the solution. Therefore, a suitable ligand is required to balance the $Al^{3+}$ complexation rate and the subsequent uniform surface deposition. DFT calculations were then performed to guide the choice of an appropriate ligand to prepare a Ni-rich oxide precursor, which can help prepare the ideal aluminium-element-modified $LiNi_{0.9}Co_{0.1}O_2$ cathode (NC91). Three typical ligands—ethanol, oxalate and ethylenediamine tetraacetate (EDTA)—were used to coordinate with $Al^{3+}$ together with $H_2O$ as the reference, and the optimised configurations for $Al^{3+}$ that was coordinated with these ligands are shown in Fig. 1f. The binding energy of $Al^{3+}$ coordinated with $H_2O$, ethanol, oxalate and EDTA was –1.67, –1.29, –7.81 and –14.64 eV with a coordination distance of 1.956, 2.029, 1.898 and 1.869 Å, respectively. The values of the charge transfer between $Al^{3+}$ and the ligand are calculated to be 0.410, 0.329, 0.416 and 0.613 e for $H_2O$, ethanol, oxalate and EDTA, respectively, which matches the binding energy results. These results show that ethanol could coordinate with $Al^{3+}$ weakly as with $H_2O$, whereas oxalate coordinates with $Al^{3+}$ strongly but is weaker than that observed with EDTA. The weak interaction between $Al^{3+}$ and $H_2O$ and ethanol leads to rapid $Al(OH)_3$ formation in solution through explosive self-nucleation with $OH^-$ addition, which makes heterogeneous nucleation and precursor surface growth difficult. In contrast, the moderately strong binding strength of oxalate towards $Al^{3+}$ slows the $Al(OH)_3$ precipitation formation rate, which is favourable for the homogeneous $Al(OH)_3$ layer coating on the Ni-based precursor surface. However, the tightest EDTA binding strength could suppress $Al(OH)_3$ formation because of an unavailable release of $Al^{3+}$. The coordinated compounds should be adsorbed on the precursor before $Al(OH)_3$ formation. Along this line, the Al-$H_2O$ and Al-oxalate coordination compound adsorption energies on the Ni-rich precursor surface were calculated. As shown in Fig. 1g, the

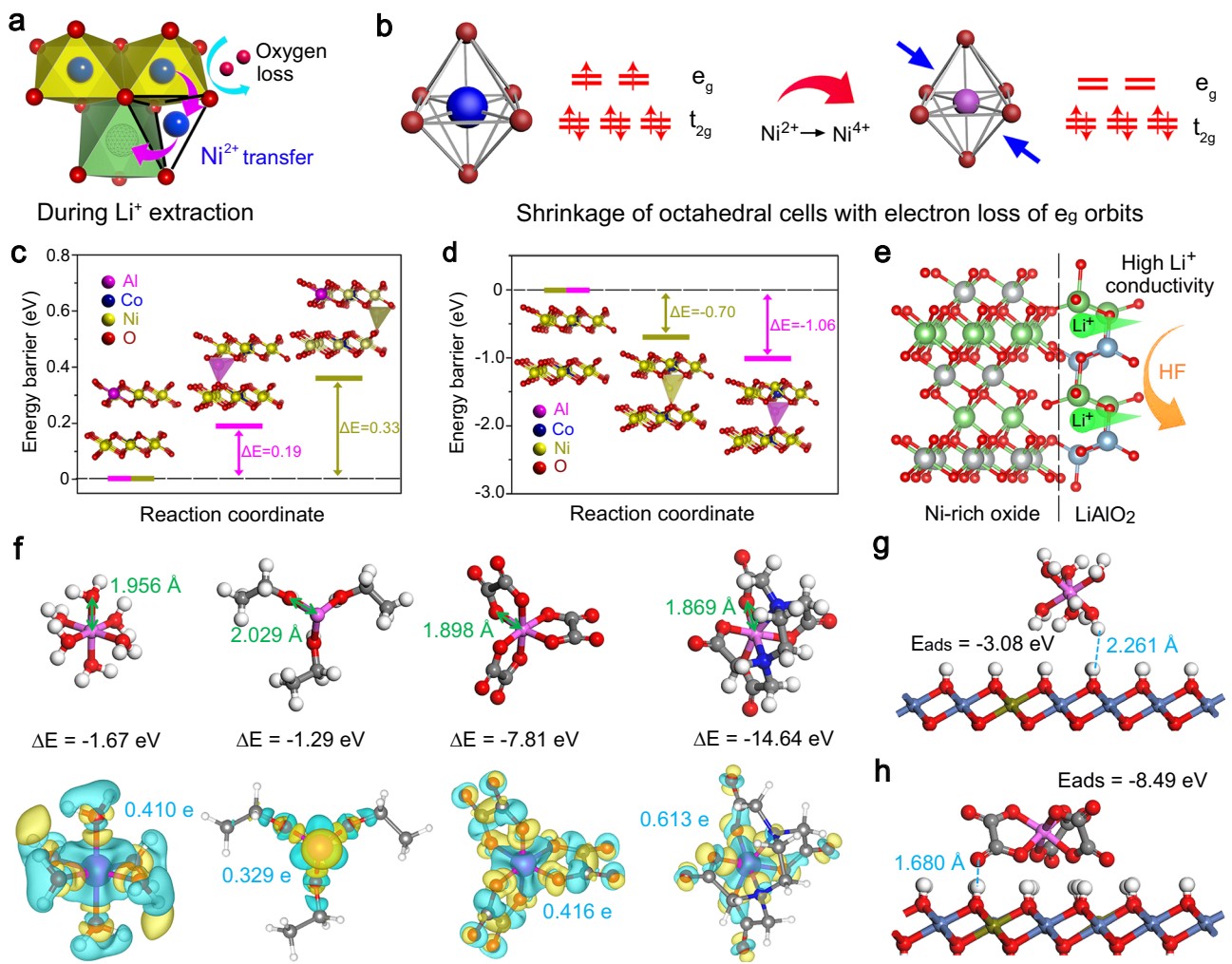

**Fig. 1 Failure mechanism and modification design of Ni-rich cathode. a** $Ni^{2+}$ transfer through tetrahedral interstice with loss of lattice oxygen after $Li^+$ extraction. **b** Shrinkage of octahedral cell of nickel ions caused by electron loss of $e_g$ orbit during oxidation. Red sphere for oxygen, blue sphere for bivalent nickel, purple sphere for tetravalent nickel, hollow sphere for Li vacancy and blue arrow for shrinkage of unit cell. **c** $Al^{3+}$ and $Ni^{2+}$ migration from octahedral interstices ($O_{TM}$) to tetrahedral interstices in the Li layer ($T_{Li}$) and corresponding energy barriers. **d** Energy barriers of $Al^{3+}$ and $Ni^{2+}$ occupying $T_{Li}$ sites. Purple sphere for aluminium, blue sphere for cobalt, yellow sphere for nickel and red sphere for oxygen. **e** Schematic of function for $LiAlO_2$ coating. Red sphere for oxygen, green sphere for lithium, grey sphere for nickel and blue sphere for aluminium. **f** Optimised configurations for $Al^{3+}$ coordinated with different ligands and corresponding charge density distributions, where the binding energy ($\Delta E$) and number of transferred charges between $Al^{3+}$ and ligands and coordination compound distance are included. Green and yellow denote 0.005 e of electron depletion and accumulation, respectively. Adsorption energies for coordination compounds of **g** $Al-H_2O$ and **h** Al-oxalate on $Ni_{0.9}Co_{0.1}(OH)_2$ surface. White, red, grey, blue and purple represent hydrogen, oxygen, carbon, nitrogen and aluminium atoms, respectively.

adsorption energy of the $Al-H_2O$ on the surface was –3.08 eV, and the distance between the $Al-H_2O$ and the surface was 2.261 Å. For Al-oxalate coordination compound, the adsorption energy increased to –8.49 eV and the distance decreased to 1.680 Å (Fig. 1h), which suggests that Al-oxalate adsorption on the precursor was more favourable. In other words, the Al-oxalate was adsorbed easily on the precursor, and $Al^{3+}$ was released moderately to ensure preferential nucleation and growth on the precursor surface during precipitation. Therefore, it can be concluded that ligands with a moderate binding strength to $Al^{3+}$, such as oxalate, are good candidates in the homogeneous coating of an $Al(OH)_3$ layer on the Ni-based precursor surface.

**Synthesis and characterisation of NCAl-LAO cathodes.** On the basis of the calculated results, an oxalate-assisted deposition protocol was proposed to modify Ni-rich oxide precursors that we

synthesised (Supplementary Fig. 3a). Figure 2a shows that the oxalate anions first chelate with $Al^{3+}$ in solution to form Al-oxalate ([Al $(C_2O_4^{2-})_3]^{3-}$) chelates. The resultant $[Al(C_2O_4^{2-})_3]^{3-}$ penetrates along intergranular gaps and adheres to exposed surfaces of the $Ni_{0.9}Co_{0.1}(OH)_2$ precursor by hydrogen-bond interaction[31,32]. With $OH^-$ introduction, the $Al^{3+}$ in the Al-oxalate chelate will be induced in a controlled manner to nucleate and grow to form an $Al(OH)_3$ coating layer on the surface (termed NC91-$Al(OH)_3$). As shown in the field emission scanning electron microscopy (FESEM) images of Supplementary Fig. 3 b, c, a uniform coating layer is visible on the NC91-$Al(OH)_3$ surface. Figure 2b provides STEM energy dispersive X-ray spectroscopy (EDS) mapping of elemental Al and Ni in cross-sectional NC91-$Al(OH)_3$, which shows that $Al(OH)_3$ is enriched mainly on the secondary particle surface, whereas a certain amount of $Al(OH)_3$ infiltrates the intergranular spaces of the primary particles. After lithiation at a high temperature, the $Al(OH)_3$ coating can be partially in situ transformed to $Li^+$-conductor

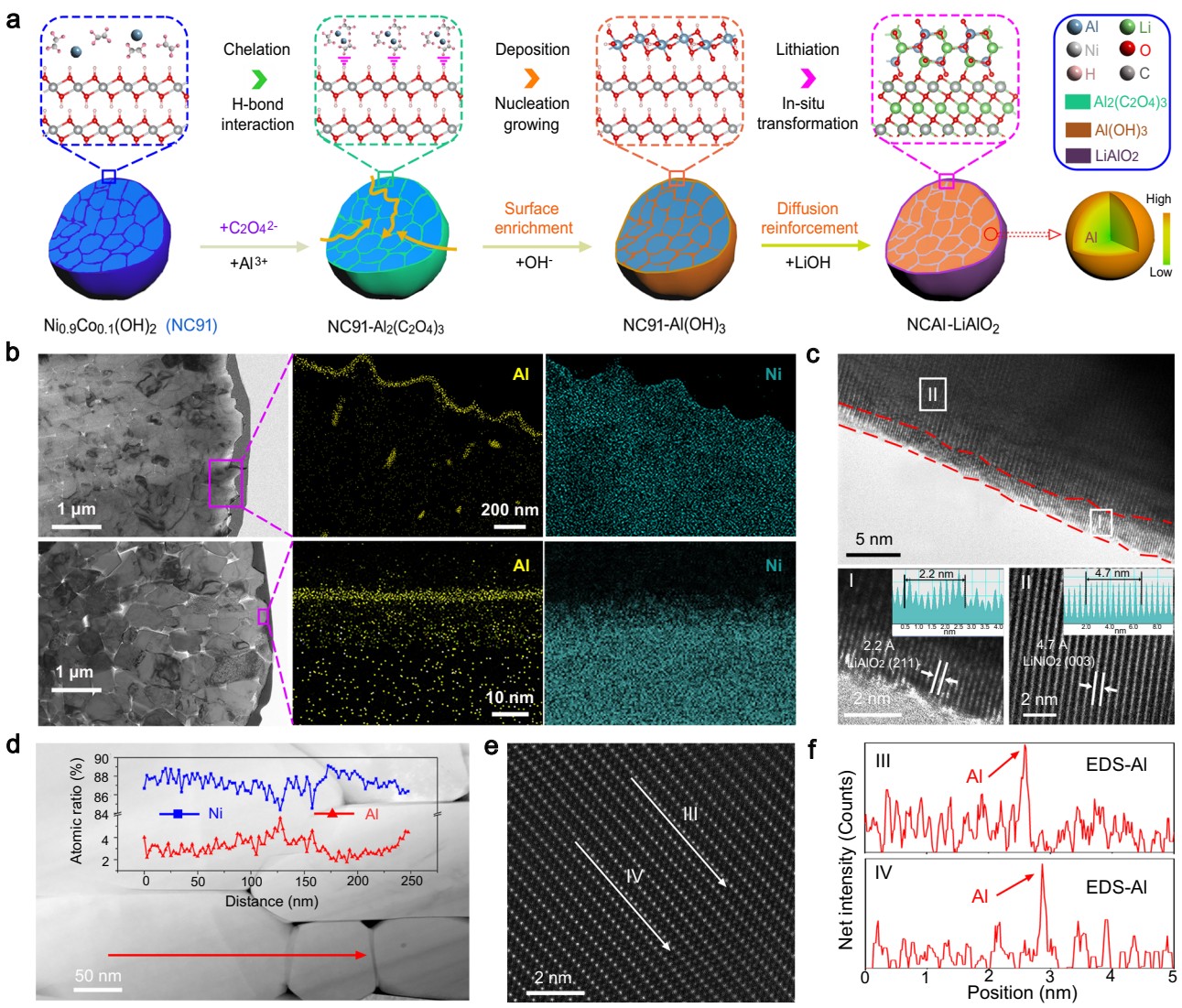

**Fig. 2 Synthesis and characterisation of aluminium-modified cathodes. a** Schematic illustration of synchronous gradient Al-doped and LiAlO$_2$-coated LiNi$_{0.9}$Co$_{0.1}$O$_2$ cathode. **b** TEM images and corresponding EDS mapping of NC91-Al(OH)$_3$ and NCAl-LAO. Yellow dot for aluminium and green dot for nickel. **c** High-resolution TEM images of surface for NCAl-LAO. **d** STEM-HAADF image of internal particles and corresponding EDS element line distribution of Ni (blue line) and Al (red line). **e**, **f** Cs-STEM-HAADF image with corresponding EDS line analysis of Al (red line) between TM layer and Li layer. Red and white arrows for EDS line scan.

LiAlO$_2$, and the gradient of Al$^{3+}$ doping into primary particles is achieved by thermally driven diffusion. The bottom half of Fig. 2b shows a homogeneous ~3.0-nm-thick coating layer on the outermost surface of the modified cathodes and other Al elements distributed in the particles (defined as NCAl-LAO). X-ray diffraction (XRD) was used to investigate the crystalline phase of the samples (Supplementary Fig. 4), which proves that the modification does not destroy the layered crystal structure. High-resolution transmission electron microscopy was used to study the detailed structural characteristic of the NCAl-LAO. In contrast with the NC91 (Supplementary Fig. 5), the outermost secondary NCAl-LAO particle surface has a homogeneous 2–4-nm-thick coating layer (Fig. 2c). The lattice spacing of 2.2 Å in the coating layer can be indexed to the (211) plane of the LiAlO$_2$ (JCPDS 38-1464, region I)[33], whereas the lattice spacing of 4.7 Å is ascribed from the (003) planes of the Ni-based layered oxides (JCPDS 09-0063, region II)[12,15].

An internal region was selected randomly in Fig. 2b and the crystal structures at the interface of the different primary particles were characterised by high-resolution transmission electron

microscopy (Supplementary Fig. 6). The distinct interplanar spacings of 4.7 and 2.0 Å in the internal particles correspond to the (003) and (104) planes of the Ni-based layered oxide[11,12], respectively, which indicates a well-preserved crystal structure after modification, and is consistent with the XRD results. No obvious lattice fringes that were ascribed to LiAlO$_2$ were observed, which suggests that aluminium compounds in the intergranular gap were thoroughly consumed. During calcination, the Al ions diffuse into the host lattice because of the driving force that is generated from the concentration difference under high temperature. Analyses of X-ray photoelectron spectroscopy (XPS) and XRD results indicated the improved dynamics and structural stability of the NCAl-LAO (Supplementary Fig. 7). To confirm the Al$^{3+}$ distribution in primary particles, a stochastic region was singled out to perform EDS line analysis in a STEM-HAADF image of the NCAl-LAO (Fig. 2d). The Al$^{3+}$ content showed a gradient distribution in a single primary particle, with the maximal content at the interface between two primary particles. XPS depth profiles by argon ion etching were applied to

reveal the Al distribution in the particle. As shown in Supplementary Fig. 8a, b, the atomic percentage of $Al^{3+}$ decreased gradually from ~7 to 2% with an increase in etching depth up to 120 nm, which proves the gradient doping of $Al^{3+}$. According to previous work[17,18], $Ni^{2+}$ cation mixing and structure disintegration firstly start from the outer surface. For the interior of particles which are more stable than the surface, a small amount of $Al^{3+}$ is capable for alleviating $Ni^{2+}$ cation mixing. EDS mapping images in the inset of Supplementary Fig. 8b show the pervasive distribution of elemental Al in the NCAl-LAO, which matches with EDS point analyses in Supplementary Fig. 8c. Therefore, a uniform surface coating on the secondary particles and gradient doping in primary particles have been manifested. The undamaged spherical morphology with an appropriate size distribution can ensure the high tap density of 2.59 g cm$^{-3}$ of the NCAl-LAO, which guarantees that the particles are applicable for high-energy density active materials (Supplementary Fig. 9)[32].

To assess the effect of Al doping on the electrochemical reaction of the Ni-rich cathode, the double aberration-corrected scanning transmission electron microscopy (Cs-STEM) with corresponding energy dispersive spectroscopy (EDS) was employed to precisely characterise the structure of the NCAl-LAO after being charged at 4.3 V. The representative Cs-STEM images acquired in high-angle annular dark-field (HAADF) mode along with the [100] orientation clearly demonstrate the well-maintained layered structure (Fig. 2e). Considering that the atom contrasts in the HAADF images depend on the atomic number, the layers with bright dots can be assigned the TM layers, and those with the dark dots can be assigned to the Li layers[34]. As schematically shown by the model (Supplementary Fig. 11a), the tetrahedral sites should be located at the gap of TM layer and Li layer. Then, Al element EDS line analysis was carried out along the white arrows between the TM layer and Li layer in Fig. 2e. The line scan profile exhibit obvious peaks corresponding to Al, which visualises the presence of $Al^{3+}$ in the tetrahedral sites at the charged state (Fig. 2f). Furthermore, the fast Fourier transform pattern for the HAADF image reveals the hexagonal phase with the R-3m space group rather than the cubic phase with the Fm-3m space group. Therefore, the presence of $Al^{3+}$ in the tetrahedral sites suppress the migration of $Ni^{2+}$ to the vacancies in the Li layers during electrochemical process. Note that only parts of nickel ions rather than all of nickel ions in Ni-based cathode materials will transfer to Li vacancies after $Li^+$ extraction, in which the part of nickel ions preferring to transfer to Li vacancies could be limited by the doped $Al^{3+}$ ions, while the other part of nickel ions that not easily transfer to Li layer can stay at the Ni layer. Therefore, the region even between two doped $Al^{3+}$ ions can aslo maintain the layered structure.

**Electrochemical performance of half and full cells.** The lithium storage properties of the NC91 and the NCAl-LAO cathodes with different aluminium contents were investigated by assembling coin-type half cells with a Li metal anode. The rate capabilities for 2.7–4.3 V for all samples are shown in Fig. 3a. The NC91, NCAl-LAO, and NCAl-LAO-L exhibit similar initial discharge capacities of ~220 mAh g$^{-1}$ at 0.2 C, whereas NCAl-LAO-H shows the lowest capacity (~200 mAh g$^{-1}$) because of the introduction of more electrochemical inert Al elements. NCAl-LAO exhibits the highest capacity of 127.7 mAh g$^{-1}$ at 20 C, and outperforms NC91 (38.6 mAh g$^{-1}$). The enhanced rate capability of NCAl-LAO is attributed to the reduced polarisation and elevated $Li^+$ transfer dynamics (Supplementary Figs. 12 and 13). To verify the repeatability of this oxalate-assisted modification method, the rate performances of NCAl-LAO prepared by different batches

(Fig. 3b) were determined. The standard deviations ($\sigma$) of the specific capacities were 0.50, 0.82, 1.27, 0.97 and 2.68 mAh g$^{-1}$ at current densities of 0.2, 1, 3, 10 and 20 C, respectively. The low $\sigma$ values within different batches and the small capacity deviations in the same batches indicate the feasibility of this deposition method. The cycle stabilities of NCAl-LAO and NC91 at 1 C are shown in Fig. 3c. NCAl-LAO retained 97.4% (194.1 mAh g$^{-1}$) of the initial capacity after 100 cycles, which is higher than that of the NC91 (85.9% with a retained capacity of 159.3 mAh g$^{-1}$). The well-maintained voltage platform and invariant phase transition during long-term operation indicate the NCAl-LAO reversibility and durability (Supplementary Fig. 14). The improved cycle performance is enhanced by the interfacial and structural stability (Supplementary Figs. 15–17), which is displayed and described in Supplementary Fig. 18.

On the basis of the electrochemical performance of the NCAl-LAO cathodes, pouch cells (capacity of ~3.6 Ah) with NCAl-LAO cathodes and commercial graphite anodes were assembled. The pouch cell (inset of Fig. 3d) had a capacity retention of 76.1% when the charge/discharge current densities increased from 0.1 (3.65 Ah) to 5 C (2.78 Ah). To confirm the high-power characteristic of this pouch cell, a rapid charging capability was examined with a constant discharge at 1 C. The pouch cell delivered a high discharge capacity of 3.37 ± 0.04 Ah (Fig. 3e) after charging at 0.5–5 C. It had the same rapid discharging ability, 3.25 ± 0.15 Ah at 0.5–5 C with constant charging at 1 C (Fig. 3f). The long-term cycle durability of the pouch cell was carried out at 1 C. As shown in Fig. 3g, the pouch cell maintained 94.4% of the initial capacity after 500 cycles with a high coulombic efficiency above 99.9%. The voltage drop was only 0.019 V with a retention of 99.5% after 500 cycles (Fig. 3h). The pouch cell delivered an energy density of ~485 Wh kg$^{-1}$ at the active material level (cathode and anode), which outperformed the commercial state-of-the-art LIBs (300 Wh kg$^{-1}$)[28,35]. Therefore, the rapid charging/discharging capabilities and superior cycle durability at a 100% depth of discharge (2.8–4.3 V) promoted the application of NCAl-LAO in dual-high type (high-energy and power-dense) LIBs with a long operation life.

**Structural investigation during electrochemical reactions.** To assess the effect of Al doping on the electrochemical reaction of the Ni-rich cathode, in situ XRD characterisation was carried out during charging to 4.3 V at 0.2 C. Figure 4a and Supplementary Fig. 19 show the (003) peak evolutions for NCAl-LAO and NC91. The (003) peaks for both cathodes shifted initially to a lower angle until ~4.0 V, which suggests $c$-axis expansion accompanied by a phase transition from H1 to H2. With further charging to 4.3 V, the (003) peaks shifted backward to a higher angle, which indicates shrinkage along the $c$-axis with phase evolution to H3[9,36]. The amplitude for $c$-axis contraction of the NCAl-LAO was smaller than that of the NC91 above 4.0 V, which implies less phase transition from H2 to H3. Contraction of the $c$-axis was related mainly to a shrinkage of the interlayer distances between the $NiO_2$ sheets because electron loss at $e_g$ orbits during oxidation from $Ni^{2+}$ to $Ni^{4+}$ caused severe shrinkage of the octahedral cell of the nickel ions[8,36]. $Ni^{2+}$ in the Li layer caused by cation mixing was oxidised to $Ni^{4+}$, generating $NiO_2$ sheets at a high charging state. These would shrink the Li layer like the TM layer and eventually cause $c$-axis contraction in the lattice, which is considered as a feature of H2–H3 phase transition[37,38]. Therefore, the effective alleviation of $Ni^{2+}$ transfer from the blocking action of $Al^{3+}$ suppressed $c$-axis shrinkage and H2–H3 phase transition, as shown by the shorter voltage plateau of the NCAl-LAO compared with the NC91 at ~4.15 V. These results are consistent with the DFT calculations on Al doping. The estimated lattice parameters

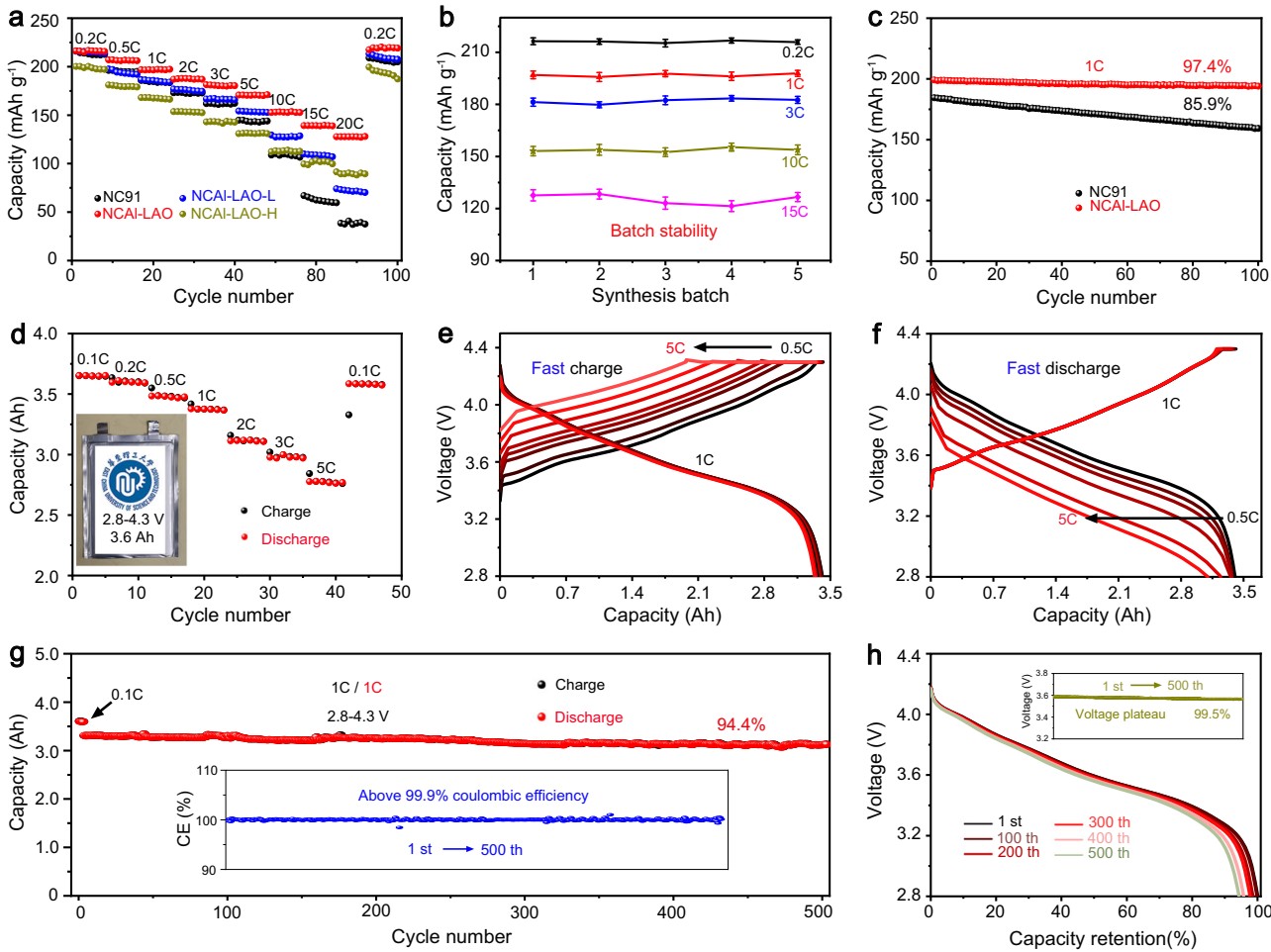

**Fig. 3 Electrochemical performances of the cathodes in half-/full cells. a** Specific capacities at 0.2–20 C of NC91 and various Al-modified NCAl-LAO for 2.7–4.3 V. **b** Performance stability data of optimised NCAl-LAO from five batches. The error bars represent the standard deviation from five independent tests. **c** Cycle performance at 1 C of NCAl-LAO and NC91. **d** Rate capability, **e**, **f** rapid charge and discharge performance, **g**, **h** cycle stability and corresponding discharge curves of NCAl-LAO/graphite full cells for 2.8–4.3 V. Inset in **d**, pouch cell with capacity of 3.6 Ah.

and unit cell volumes from the in situ XRD refinements as a function of the charging state are plotted in Fig. 4b, c. For the NCAl-LAO and NC91, there was almost no difference in change magnitude of the above two parameters at the early stage of charge (<4.0 V). However, an obvious distinction occurred between the two samples during subsequent charging (>4.0 V). The maximum shrinkage (Δc) for NC91 (4.7%) was much higher than that of NCAl-LAO (2.1%). For a unit cell volume, the maximum change value (ΔV) was 6.9% for NC91, which is higher than the 3.8% for NCAl-LAO.

Cross-sectional SEM images of the electrodes after 100 cycles are shown in the inset of Fig. 4b, c. Secondary NCAl-LAO particles remained mostly free from cracks, whereas the NC91 contained large cracks that traversed entire particles and segment secondary particles into independent fragments. The contraction of lattice along the c-axis and the shrinkage of unit cell volume will cause anisotropic volume change of the primary particles to generate internal mechanical stresses, which is considered as the origin of crack formation. Therefore, the volume deformation and stress were simulated at different charge states to validate the reinforced mechanical stability of the NCAl-LAO, and the corresponding model and change of lattice parameter are shown in Supplementary Figs. 20 and 21. At 4.0 V, the volume deformations and internal stresses of both cathodes were approximate because the changes in lattice parameters were

similar at the early stage of charge (Supplementary Fig. 22). The distributions of volume deformation and internal stresses inside the secondary particles at 4.3 V are shown in Fig. 4d. Because of the stochastic crystallographic orientations, different primary particles were subjected to different types of volume deformation, including tensile and shrinkage, which was reflected by positive and negative values, respectively. The NCAl-LAO exhibited a lower volume deformation in any region, which occurred mainly because of its smaller c-axis and unit cell volume shrinkage at the end of charging. The corresponding Von mises equivalent stresses from Hook's law are shown in Supplementary Fig. 23a, c. NCAl-LAO bore a smaller equivalent stress with an average value of 269 MPa, whereas that of NC91 was as high as 357 MPa. The equivalent stresses were subdivided into tensile (represented by positive values) and compressive (represented by negative values) stresses, which reflects the true stress situation and clarifies the adverse effects of these internal stresses (Supplementary Fig. 23b, d). The lighter colours correspond to tensile and compressive stresses and indicate that the stresses are smaller at every region inside NCAl-LAO. The standard deviations of the tensile and compressive stresses for NCAl-LAO were 75 and 77 MPa, respectively, which are lower than those of NC91 (7666 and 113 MPa). A larger standard deviation suggests that an abnormally high stress occurs in the local area in NC91, such as an interface of the primary particles (enlarged image in

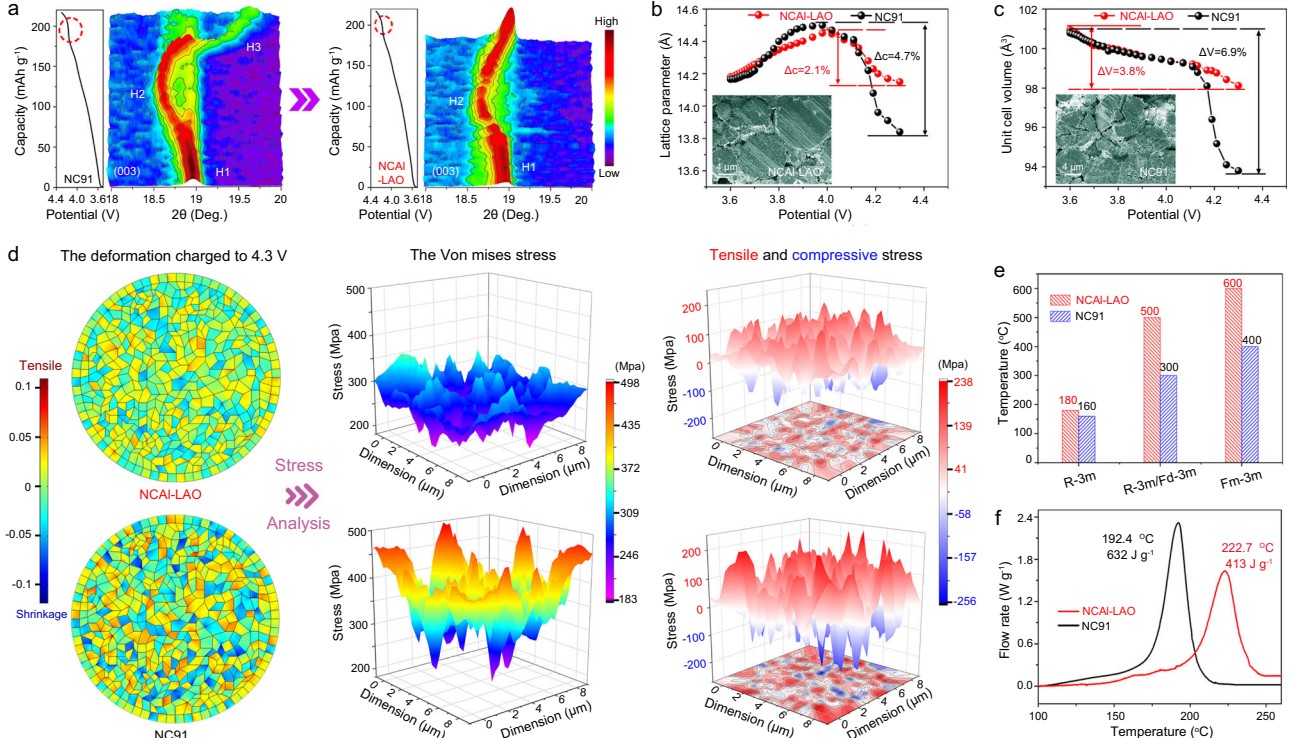

**Fig. 4 Crystal structure, internal stress, and thermal stability of Ni-rich cathode. a** Three-dimensional contour plots of (003) peaks from in situ XRD patterns for NC91 and NCAl-LAO with corresponding capacity vs. potential curves. Variations of **b** c-axis lattice parameters and **c** unit cell volume as a function of charging potential, and insets show cross-sectional SEM images of electrodes after 100 cycles. **d** Distributions of volume deformation, corresponding Von mises stress, tensile and compressive stresses through NCAl-LAO and NC91 when charging to 4.3 V. **e** Phase transformation temperature of NCAl-LAO and NC91 from HT-XRD patterns. **f** Differential scanning calorimetry profiles of NCAl-LAO and NC91 in delithiated state (charged to 4.3 V).

Supplementary Fig. 23b), which will tend to cause microcrack nucleation and destroy the mechanical integrity of secondary particles. In conclusion, suppressed $Ni^{2+}$ transfer in NCAl-LAO mitigates c-axis and unit cell volume shrinkage accompanied by H2–H3 phase transition alleviation and a reduction in volume deformation and internal stress inside particles, which boosts the structural integrity and robustness.

**Thermal stability of NCAl-LAO.** In situ high-temperature XRD (HT-XRD) characterisation was carried out to assess the structural and thermal stabilities of the delithiated NC91 and NCAl-LAO. As shown in Fig. 4e and Supplementary Fig. 24, the initial rhombohedral structures (space group of R-3m) were maintained after charging to 4.3 V. With an increase in temperature, the NCAl-LAO transformed to a disordered spinel structure (space group of Fd-3m) at ~200 °C, as shown by the coalescence of $(108)_R$ and $(110)_R$ peaks into a single spinel $(440)_S$ peak[39]. This was followed by another phase transformation until all main peaks were well indexed to a cubic rock salt phase (space group of Fm-3m) without other phases at ~600 °C[40]. For NC91, the corresponding transition temperatures of the disordered spinel and rock salt phases decreased to ~180 °C and 400 °C, respectively. The postponed phase transformation attests to the enhanced structural and thermal stabilities of the NCAl-LAO. The differential scanning calorimetry (DSC) tests (Fig. 4f) showed that the NC91 had a sharp exothermic peak at 192.4 °C, with a total released heat of 632 J g$^{-1}$. In comparison, the NCAl-LAO delivered a higher exothermic peak temperature of 222.7 °C and a lower total released heat of 413 J g$^{-1}$, which indicates the reinforced thermal stability.

## Discussion

Synchronous surface-enrichment gradient Al-doped and $LiAlO_2$-coated $LiNi_{0.9}Co_{0.1}O_2$ cathodes (NCAl-LAO) were synthesised. Theoretical calculations, HT-XRD and in situ XRD results showed that $Al^{3+}$ mitigated $Ni^{2+}$ transfer to the Li layer, which enhanced the thermal stability and suppressed the H2–H3 phase transition of the Ni-rich cathode. Finite element analysis confirmed the mitigated volume deformation and internal stress. The $LiAlO_2$ coating and inhibition of structure crack decreased side reactions with the electrolytes. The as-prepared NCAl-LAO exhibited a high capacity and excellent capacity retention after cycling. The pouch cell that was assembled by NCAl-LAO and the graphite anode displayed a stable cycle life over 500 cycles with an unprecedented rapid charging/discharging performance. This work provides a scalable strategy to achieve high-energy and power-dense Ni-rich cathodes with a long life and good thermal stability.

## Methods

**Materials preparation.** Spherical $Ni_{0.90}Co_{0.10}(OH)_2$ precursors were firstly prepared via a co-precipitation method using a continuous stirred tank reactor (5 l). Briefly, the aqueous solution of $NiSO_4 \cdot 6H_2O$ and $CoSO_4 \cdot 7H_2O$ (Ni:Co = 9:1, 2 M), 4 M NaOH (aq.) and 1.8 M $NH_4OH$ (aq.) were simultaneously dripped into the reactor containing a base solution of $NH_4OH$ (0.6 M) under an argon atmosphere, and the feed rate was carefully controlled to maintain a constant pH (pH = 11.2). After reacting for 20 h, the synthesised precursors were washed with deionized water, filtered and then dried at 120 °C overnight. The $Al(OH)_3$ coated precursors were prepared by an oxalate-assisted deposition strategy. Typically, $Al_2(SO_4)_3 \cdot 18H_2O$ and ammonium oxalate mixing solution was pumped into the dispersion liquid of $Ni_{0.90}Co_{0.10}(OH)_2$. Subsequently, the $NH_4OH$ (aq.) was added to aforementioned suspension liquid and maintain the pH of reaction solution at 9 by adjusting the flow rate. After stirring for 4 h, the suspension liquid was filtrated and then the $Al(OH)_3$ coated $Ni_{0.90}Co_{0.10}(OH)_2$ precursors (NC91-Al(OH)$_3$) were obtained after drying at 120 °C for 12 h. Finally, NC91-Al(OH)$_3$ precursors were

mixed with $LiOH \cdot H_2O$ (molar ratio of lithium to TM = 1.05) and preheated at 500 °C for 5 h, then sintered at 720 °C for 12 h under a pure $O_2$ atmosphere to obtain the modified cathodes. To select the optimal sample, the modified cathode with low, moderate and high Al contents were prepared, and the cathode were denoted as NCAl-LAO-L, NCAl-LAO and NCAl-LAO-H, respectively. The $LiNi_{0.9}Co_{0.1}O_2$ (NC91) was also obtained by using pristine $Ni_{0.90}Co_{0.10}(OH)_2$ precursors.

**Materials characterisation.** The chemical compositions of all samples were analysed by the inductively coupled plasma atomic emission spectrometer (Agilent 725). The crystalline phases of the powders were characterised by XRD (Bruker D8 Advance) and the corresponding lattice parameters were calculated by Rietveld refinement. To assess the structural stability, the delithiated cathodes were tested by in situ HT-XRD (30–600 °C) with temperature increase from 30 to 600 °C. In situ XRD was carried out by Rigaku Ultima IV connecting with a LANDCT2001A test system to monitor the phase transition during charging process, and the modified coin-type half-cells were gradually charged to 4.3 V at a constant current of 40 mA $g^{-1}$ with recording the diffraction patterns around every 3.5 min. The morphologies were observed by FESEM (GeminiSEM 500). The microstructure and element distribution were analysed via high-resolution transmission electron microscopy (FEI Talos F200X) and double aberration-Cs-STEM (FEI Themis Z) with accessory of EDS, and the part of the samples were prepared by focused ion beam etching technique (TESCAN GALA 3). The surface chemical valence states of the relevant elements were analysed by XPS (ESCA PHI500C). The thermal stability was measured by DSC (NETZSCH DSC204) using charged cathode with existence of electrolyte.

**Electrochemical test.** The electrochemical performances were measured via a coin-type 2016 cell. The active materials were homogenised with super P and poly (vinylidene fluoride) at a mass ratio of 8:1:1 in N-methyl pyrrolidone. After stirring for 4 h, the homogeneous slurry was coated on pure aluminium foil and drying for 12 h at 120 °C in absolute vacuum. The half-cells were assembled in an argon-filled glovebox with pure lithium metal as the counter electrode, which were separated by a polypropylene membrane (Celgard-2400). For the pouch cells, the NCAl-LAO electrode with high loading mass of ∼26 mg $cm^{-2}$ was matched with commercial graphite via suitable capacity ratio (1:1.1), and all of electrodes were doubled coated. Thickness and porosity of the elctrodes employed in the final pouch cell are 113 μm and 42%, respectively. After welding the pole ears and slitting (57 × 11.5 $cm^2$), the cathode, anode and separator were stacked in sequence and winded to obtain the square core of LIBs, which was then packed by Al-plastic film and injected electrolyte in an argon-filled glovebox. Ultimately, a square pouch full cell with capacity of ∼3.6 Ah was obtained after per charge and degassing processes. The electrolyte was 1.2 M $LiPF_6$ dissolved in a mixture of ethylene carbonate and ethyl methyl carbonate (3:7 by volume) with 2 wt% VC, and amounts of electrolyte used in coin cell and in pouch cell are 100 μl and 8 g, respectively. The galvanostatic charge and discharge measurements were performed with a LANDCT2001A test system within a voltage range of 2.7–4.3 V at various current densities, and the value of 1 C was 180 mA $g^{-1}$. Cyclic voltammetry curves at sweep rate of 0.2–1 mV $s^{-1}$ and electrochemical impedance spectra at constant voltage mode (100 kHz–0.01 Hz) were executed in an Autolab PGSTAT302N electrochemical workstation. For the galvanostatic intermittent titration technique characterisation, the batteries were discharged repeatedly at 0.1 C for 20 min, which then lasts for 2 h at the open-circuit up until 2.7 V.

**Theoretical calculations.** The migration and formation energies of $Al^{3+}$ and $Ni^{2+}$ between octahedral and tetrahedral interstices in delithiated cathode were calculated, which were performed within the framework of first-principles plane-wave pseudopotential formulation as implemented in the Vienna ab initio Simulation Package[41–43]. The Perdew–Burke–Ernzerhof[44] with the generalised gradient approximation (GGA) was adopted to describe exchange-correlation functional. The cutoff energy of 500 eV for the plane-wave basis and Γ-centred k-mesh of 4 × 4 × 2 for the structural optimisation were applied, which will ensure the energy convergence was lower than 0.01 meV and the residual force acting on each atom was < 0.03 eV/Å. Considering strongly correlated d electrons in TMs, a Hubbard U correction (GGA + U) was included with U parameters of 6.5 and 4.9 eV for Ni and Co ions, respectively. The 3 × 3 × 1 supercell with total 108 atoms was used to model the solid-solution compound of $LiNi_{0.9}Co_{0.1}O_2$ by substituted three Ni with Co atoms in a configuration, in which each TM-O layer had the same ratio of Ni: Co. The formation energy (ΔE) was defined as $\Delta E = E_{doped\ system} + \mu_{substituted\ atom} - E_{pristine\ system} - \mu_{dope\ atom}$, where $E_{doped\ system}$ and $E_{pristine\ system}$ were the energy of doped and pristine $LiNi_{0.9}Co_{0.1}O_2$, respectively. $\mu_{substituted\ atom}$ and $\mu_{dope\ atom}$ were the chemical potentials of substituted and doping elements, respectively. The migration energy (ΔE) was defined as $\Delta E = E_{migrated\ system} - E_{pristine\ system}$, where $E_{migrated\ system}$ and $E_{pristine\ system}$ were the energy before and after migration, respectively. The formation energy and migration energy were all calculated with respect to formula unit (f.u.). The configurations of Al coordinated with the ligands were optimised in box of 20 × 20 × 20 Å, based on which the binding energies for Al coordinated with the ligands are calculated by $\Delta E = E_{Al-ligand} - E_{Al} - CN \times E_{ligand}$. The $E_{Al-ligand}$, $E_{Al}$ and $E_{ligand}$ are the calculated energies of Al coordinated with the ligands, free Al and ligand, respectively. CN is the coordination number for Al and the ligand, i.e., 6 for water, 3 for ethanol, 4 for oxalate and 1 for EDTA. It should be noted that the ligand of oxalate is $C_2O_4^{2-}$ rather than the neutral oxalate molecule. All the ligands are six-fold coordinated with Al except ethanol. For ethanol, the optimised results show that the most thermodynamically stable coordination is three-fold coordinated configurations. Bader analysis[45] was used to calculate electronic charges on atoms in order to identify electronic interaction between $Al^{3+}$ and ligand. The charge density difference (Δρ) isosurface is calculated by $\Delta \rho = \Delta \rho_{Al-ligand} - \Delta \rho_{Al} - \Delta \rho_{ligand}$, where $\Delta \rho_{Al-ligand}$ is the charge density of the optimised Al coordinated with ligand, $\Delta \rho_{Al}$ is the charge density of Al atoms in the corresponding location and $\Delta \rho_{ligand}$ is the charge density of the ligand in the corresponding location. The adsorption energy for Al-oxalate and Al-$H_2O$ on the surface of $Ni_{0.9}Co_{0.1}(OH)_2$, which was modelled by substituting two Ni atoms with Co atoms in the top surface of $Ni(OH)_2$ (001) using a p(3 × 3) supercell, were calculated by $\Delta E = E_{adsorbate/surface} - E_{surface} - E_{adsorbate}$. The $E_{adsorbate/surface}$, $E_{surface}$ and $E_{adsorbate}$ are the energies of the optimised adsorbate on the $Ni_{0.9}Co_{0.1}(OH)_2$ surface, the optimised surface and the free adsorbate, respectively.

## Data availability

The data that support the findings of this study are available from the corresponding author upon reasonable request.

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

## Acknowledgements
This work was supported by the National Natural Science Foundation of China (21838003, 91834301 and 51621002), the Innovation Program of Shanghai Municipal Education Commission, the National Program for Support of Top-Notch Young Professionals, and the Fundamental Research Funds for the Central Universities (222201718002).

## Author contributions
H.Y. and H.J. conceived the concept and experiments. H.Y., S.D. and H.J. performed the experiments. Y.C. and X.D. performed the computational studies. L.C., Y.H. and C.L. contributed to data analysis and manuscript editing. H.Y. and H.J. co-wrote the paper. All authors discussed the results and commented on the manuscript.

## Competing interests
The authors declare no competing interests.
