## [Peer Review File · Nature Communications]

REVIEWER COMMENTS

Reviewer #1 (Remarks to the Author):

In this manuscript, Yu et al. present a study on Al-doped and LiAlO₂-coated LiNi_{0.9}Co_{0.1}O₂ cathodes combining different experimental techniques with density functional theory (DFT) calculations. Before recommending this paper for its publication in Nature Communications, I suggest moving the DFT calculation details from the supplementary material to the manuscript's main body. In addition, I would like the authors to clarify the following point regarding their DFT calculations:

-How are the binding energies in Fig. 1f calculated? Are they given per ligand (i.e., there are 6 water molecules, 3 ethanol, 4 oxalates, and one EDTA? I think it is misleading if the binding energies are given per ligand. They should be given per the whole solvation shell. Also, all the complexes are 6-fold coordinated, but ethanol, which is 3-fold coordinated. Is it not possible to add 3 more ethanol molecules to the solvation shell of Al³⁺? What is the size of the supercell used in these calculations? What is the reference for the oxalates, neutral or C₂O₄²⁻?

Reviewer #2 (Remarks to the Author):

The paper here reported focuses on the design and investigation of synchronous gradient Al-doped and LiAlO₂-coated LiNi_{0.9}Co_{0.1}O₂ cathode prepared by using a new oxalate-assisted deposition followed by a thermally driven diffusion method.

Authors present in situ X-ray diffraction results and finite-element simulations providing insights into the Al³⁺ migration into tetrahedral interstices prior to Ni²⁺, thus eliminating Li/Ni disorder effects and internal structure stress. The Li⁺-conductive LiAlO₂ coating skin prevents deleterious side reactions with the electrolytes and enables the achievement of improved cycling stability and fast charging ability. Finally, authors provide results of a 3.5-Ah pouch cell employing the developed cathode and a graphite anode showing more than 500-long cycle life with only a 5.6% capacity loss.

The study and the characterization tools employed make the paper interesting to a broad readership. The scientific outcome is very interesting for basic science and application-driven research.

The theoretical calculations and the in-situ X-ray diffraction (at room and high temperature) study have been carefully undertaken and discussed by the authors providing insights into the structural stability (Al doping effect) and the thermal stability of the NCAI-LAO (including also DSC analysis).

The overall study is very well presented, and several consistent measurements confirm the conclusion of the authors. In addition, authors are able to provide concrete improvements in terms of cyclability of the electrode, proved at a "real-scale" cell prototype, confirming the reproducibility of the synthesis method and also the electrochemical properties obtained by different batches.

Overall, I suggest that the manuscript can be accepted for publication in Nature Comm, after addressing the comments below.

1. In other papers (e.g. ACS Appl. Energy Mater. 2019, 2, 5, 3098–3113 "Tunable LiAlO₂/Al₂O₃ Coating through a Wet-Chemical Method To Improve Cycle Stability of Nano-LiCoO₂"), the positive effect of Al-based coating has already been provided. Can authors comment in the introduction part and add more info strengthening the novelty message of this paper compared to other ones, such as the above mentioned or the refs cited by the authors 9ref 11-160?
2. Page 6: authors claim that lithium aluminum oxide (LiAlO₂) coating has an excellent Li⁺ surface conductivity can stabilize the cathode-electrolyte interface because of its high chemical stability. Have the authors measure the chemical stability? Any support for this statement? How can the authors

prove the chemical stability? In which specific environment? If this cannot be done, is there any specific reference authors can add to support the statement?

3. On the stability of the coating layer: The XPS surface analysis was performed for NCAI-LAO and NC91 after 100 cycles at 1 C to certify the stability of interfacial chemistry. The C 1s and f 1s region are showed. I would appreciated if the authors could add information of Al 2p region at XPS. The question is related to the possible fluorination of the layer. Is this occurring? Any chance the LiAlO₂ layer can be fluorinated? If yes, is this beneficial or deleterious?

4. Impedance data in Fig. 13 in Supp Info: EIS data should always be presented in squared graphs. Please, reshape the graphs.

5. Authors should add some information in the experimental part such as:

- Value of 1C considered.
- Amount of electrolyte used in coin cell and in the final pouch cell.
- Thickness and porosity of the electrodes employed in the final pouch cell.

Overall, I particularly enjoyed reading this manuscript and I believe the results reported might be beneficial to a broad audience.

Reviewer #3 (Remarks to the Author):

This manuscript demonstrates a gradient Al-doped and LiAlO₂-coated high-nickel cathode by using a new deposition method with oxalate ligands. The authors used calculations to show that Al³⁺ preferentially moves to the tetrahedral interstices compared with Ni²⁺ which eliminates cation mixing. Electrochemical performance of the coated cathode showed superb cyclability and improved power density. However, migration of Al³⁺ to the tetrahedral site has been suggested before for several times (Chem. Mater. 2003, 15, 4476-4483, J. Power Sources, 2003, 115, 305-314). As authors' suggestion even requires more obvious experimental data, I don't feel that this work is a substantial progress to be published in Nature Communications as compared to the above prior arts. The following points are encouraged to address:

1. The authors showed that Al³⁺ prefers to migrate to tetrahedral interstices in Li layer by calculations. However, it is required to add more experimental evidence, such as TEM or STEM images with FFT pattern to visualize that Al³⁺ ions truly exist in the tetrahedral sites at the charged state and block Ni²⁺ migration to vacancies in the Li layer.

2. Oxalate-mediated co-precipitation method, one of the novelties in this manuscript, has been reported before on cathode synthesis (J. Mater. Chem. A, 2015, 3, 9427, J. Mater. Chem. A, 2015, 3, 21219) with a strength that the anion balances chelating level of metal ions. Though the authors showed the propriety of using oxalate ligand for Al coating by calculations, the novelty of the method can't help being discolored.

3. In the electrochemical performance part, the coated cathodes showed small surface film resistance and reduced polarization, while the authors gave no demonstration why the kinetic properties of the cathodes were improved. It is generally reasonable that the impedance of the cathodes increases when coated. Please explain why.

4. According to in-situ XRD data coating and gradient doping helps to reduce H2-H3 transition and suppress microcracks. However, there is no clear correlation between cation mixing and H2-H3 transition. Besides, it is still faint how surface coating and doping can even control the bulk phenomena such as H2-H3 transition and microcrack formation. In addition, according to Fig. 2f and

Supplementary Fig. 6, Al³⁺ rarely exists more than 60 nm away from the grain boundaries. In the region without Al³⁺, it is wondering how Ni²⁺ cation mixing can be eliminated.

To Reviewer #1:

Comments: In this manuscript, Yu et al. present a study on Al-doped and LiAlO₂-coated LiNi_{0.9}Co_{0.1}O₂ cathodes combining different experimental techniques with density functional theory (DFT) calculations. Before recommending this paper for its publication in Nature Communications, I suggest moving the DFT calculation details from the supplementary material to the manuscript's main body. In addition, I would like the authors to clarify the following point regarding their DFT calculations:

Reply: Really appreciate the constructive and probing comments from the reviewer. In the revision, we have moved the details for DFT calculations from the supplementary material to the main body of the revised manuscript.

1. How are the binding energies in Fig. 1f calculated? Are they given per ligand (i.e., there are 6 water molecules, 3 ethanol, 4 oxalates, and one EDTA)? I think it is misleading if the binding energies are given per ligand. They should be given per the whole solvation shell.

Reply: Many thanks for the reviewer's kind questions. The binding energies for Al³⁺ coordinated with the ligands are calculated by $\Delta E = E_{\text{Al-ligand}} - E_{\text{Al}} - E_{\text{ligand}}$. $E_{\text{Al-ligand}}$, E_{Al} and E_{ligand} are the calculated energies of Al coordinated with the ligands, free Al³⁺ and ligand, respectively. It should be noted that the energy of the free ligands is the energy of single ligand multiplied by the coordination number (i.e., 6 for water molecules, 3 for ethanol, 4 for oxalates, and one for EDTA) rather than the energy of single ligand. In other words, the binding energies are given per the whole solvation shell. To make the readers clearer, we have modified the equation for calculating the binding energies as follows:

“The configurations of Al coordinated with the ligands were optimized in box of 20 Å × 20 Å × 20 Å, based on which the binding energies for Al coordinated with the ligands are calculated by $\Delta E = E_{\text{Al-ligand}} - E_{\text{Al}} - \text{CN} \times E_{\text{ligand}}$. The $E_{\text{Al-ligand}}$, E_{Al} and E_{ligand} are the calculated energies of Al coordinated with the ligands, free Al and ligand, respectively. CN is the coordination number for Al and the ligand, i.e., 6 for water, 3 for ethanol, 4 for oxalate, and one for EDTA.” (Page 22, Line 16-21)

2. Also, all the complexes are 6-fold coordinated, but ethanol, which is 3-fold coordinated. Is it not possible to add 3 more ethanol molecules to the solvation shell of Al³⁺?

Reply: We faithfully appreciate the valuable questions. The 3-coordinated complex with ethanol as the ligand is the thermodynamically stable one. We ever tried to add one more molecule to coordinate with Al³⁺, but the calculated results show that the fourth ethanol cannot complexed with Al³⁺ via Al-O coordination. Instead, the hydrogen atom of the hydroxyl in the fourth ethanol decomposed and moved to bind with Al³⁺. Thus, we employed 3-coordinated ethanol molecules for the complex of Al-ethanol. To make this information clearer, we have added the relevant detail in the revised manuscript as follows:

“All the ligands are 6-fold coordinated with Al except ethanol. For ethanol, the optimized results show that the most thermodynamically stable coordination is 3-fold coordinated configurations.” (Page 22, Line 22 to Page 23, Line 2)

3. What is the size of the supercell used in these calculations? What is the reference for the oxalates, neutral or C₂O₄²⁻?

Reply: We are sorry for the absence of these details. The sizes of the supercells used in the calculations for the complex of Al coordinated with all the ligands are $20 \text{ \AA} \times 20 \text{ \AA} \times 20 \text{ \AA}$. For the adsorption of Al-oxalate and Al-H₂O on the surface of Ni_{0.9}Co_{0.1}(OH)₂, a $p(3 \times 3)$ supercell was employed for the surface. In addition, the reference for the oxalate are C₂O₄²⁻, in which two hydrogen atoms were moved away. In the revised manuscript, we have added the relevant details as follows:

“The configurations of Al coordinated with the ligands were optimized in box of $20 \text{ \AA} \times 20 \text{ \AA} \times 20 \text{ \AA}$, based on which the binding energies for Al coordinated with the ligands are calculated by $\Delta E = E_{Al\text{-ligand}} - E_{Al} - CN \times E_{\text{ligand}}$. The $E_{Al\text{-ligand}}$, E_{Al} and E_{ligand} are the calculated energies of Al coordinated with the ligands, free Al and ligand, respectively. CN is the coordination number for Al and the ligand, i.e., 6 for water, 3 for ethanol, 4 for oxalate, and one for EDTA. It should be noted that the ligand of oxalate is C₂O₄²⁻ rather than the neutral oxalate molecule.” (Page 22, Line 16-22)

“The adsorption energy for Al-oxalate and Al-H₂O on the surface of Ni_{0.9}Co_{0.1}(OH)₂, which was modeled by substituting two Ni atoms with Co atoms in the top surface of Ni(OH)₂ (001) using a $p(3 \times 3)$ supercell, were calculated by $\Delta E = E_{\text{adsorbate/surface}} - E_{\text{surface}} - E_{\text{adsorbate}}$.” (Page 23, Line 7-10)

To Reviewer #2:

Comments: *The paper here reported focuses on the design and investigation of synchronous gradient Al-doped and LiAlO₂-coated LiNi_{0.9}Co_{0.1}O₂ cathode prepared by using a new oxalate-assisted deposition followed by a thermally driven diffusion method.*

Authors present in situ X-ray diffraction results and finite-element simulations providing insights into the Al³⁺ migration into tetrahedral interstices prior to Ni²⁺, thus eliminating Li/Ni disorder effects and internal structure stress. The Li⁺-conductive LiAlO₂ coating skin prevents deleterious side reactions with the electrolytes and enables the achievement of improved cycling stability and fast charging ability. Finally, authors provide results of a 3.5-Ah pouch cell employing the developed cathode and a graphite anode showing more than 500-long cycle life with only a 5.6% capacity loss.

The study and the characterization tools employed make the paper interesting to a broad readership. The scientific outcome is very interesting for basic science and application-driven research.

The theoretical calculations and the in-situ X-ray diffraction (at room and high temperature) study have been carefully undertaken and discussed by the authors providing insights into the structural stability (Al doping effect) and the thermal stability of the NCAI-LAO (including also DSC analysis).

The overall study is very well presented, and several consistent measurements confirm the conclusion of the authors. In addition, authors are able to provide concrete improvements in terms of cyclability of the electrode, proved at a “real-scale” cell prototype, confirming the reproducibility of the synthesis method and also the electrochemical properties obtained by different batches.

Overall, I suggest that the manuscript can be accepted for publication in Nature Comm. after addressing the comments below.

Reply: We highly appreciate these positive comments from the reviewer.

1. In other papers (e.g. ACS Appl. Energy Mater. 2019, 2, 5, 3098-3113 “Tunable LiAlO₂/Al₂O₃ Coating through a Wet-Chemical Method to Improve Cycle Stability of Nano-LiCoO₂”), the positive effect of Al -based coating has already been provided. Can authors comment in the introduction part and add more info strengthening the novelty message of this paper compared to other ones, such as the above mentioned or the refs cited by the authors ref 11-16?

Reply: Many thanks for the valuable suggestions. In the revised version, we have compared our results with previously studies in the section of Introduction to highlight the novelty of this present work as follows:

“Compared to previously reported single doping or coating modification^{9-16,23}, the simultaneously obtained gradient Al-doping inside the primary particles and uniform LiAlO₂ coating on the surface of the secondary particles can concurrently stabilise crystal structure and hinder the parasitic reaction at the interface. This strategy was revealed to minimize the capacity sacrifice due to the incorporation of electrochemical inert element.” (Page 3, Line 18 to Page 4, Line 1)

2. Page 6: authors claim that lithium aluminum oxide (LiAlO₂) coating has an excellent Li⁺ surface conductivity can stabilize the cathode-electrolyte interface because of its high chemical

stability. Have the authors measure the chemical stability? Any support for this statement? How can the authors prove the chemical stability? In which specific environment? If this cannot be done, is there any specific reference authors can add to support the statement?

Reply: Thanks for the kind questions. The high chemical stability mentioned in this paper means that LiAlO₂ will not dissolve in the specific environment of organic electrolyte, thus it can act as a physical barrier to protect active materials from HF corrosion. To check such stability, we further carried out an experiment, in which LiAlO₂ was immersed within the electrolyte for one week. Then, the amount of Al in the supernatant was determined by ICP-OES, which shows no dissolved Al element from the LiAlO₂. This result is also in consistent with the previous works (*J. Am. Chem. Soc.* **133**, 14741-14754 (2011); *Chem. Mater.* **26**, 3128-3134 (2014)). In the revision, we have added relevant discussion and references as follows:

“Besides the doping of Al³⁺ in the crystal structure of Ni-rich cathodes, a lithium aluminium oxide (LiAlO₂) coating with an excellent Li⁺ surface conductivity can stabilise the cathode-electrolyte interface in organic electrolyte (Fig. 1e), due to the high chemical stability of LiAlO₂ (Supplementary Fig. 2)^{29, 30}” (Page 6, Line 3-6)

“To check the chemical stability of LiAlO₂ in organic electrolyte, 200 mg of LiAlO₂ was immersed within 3 mL of electrolyte for one week, and the amount of Al in the supernatant was determined by ICP-OES. No dissolved Al element was detected by ICP-OES, which is matched well with clear and transparent supernatant in digital photograph (Supplementary Fig. 2)” (Page 5 in Supplementary information)

3. On the stability of the coating layer: The XPS surface analysis was performed for NCAI-LAO and NC91 after 100 cycles at 1 C to certify the stability of interfacial chemistry. The C 1s and F 1s region are showed. I would appreciate if the authors could add information of Al 2p region at XPS. The question is related to the possible fluorination of the layer. Is this occurring? Any chance the LiAlO₂ layer can be fluorinated? If yes, is this beneficial or deleterious?

Reply: Thanks for the constructive suggestions. We have deconvoluted the spectrum of F 1s region of NCAI-LAO after cycling (Supplementary Fig. 15), and the results show obvious Al-F signal (ca 686.5 eV) in the spectrum. In addition, we have added the spectrum of Al 2p region of NCAI-LAO after cycling (Supplementary Fig. 16), in which the obvious Al-F signal (ca 76.5 eV) was also clearly observed. These results indicate that the LiAlO₂ layer can be partly fluorinated to form AlF₃ during electrochemical process. A similar phenomenon was also observed when using another Al-containing coating (*J. Phys. Chem. C* **111**, 4061-4067 (2007); *Chem. Mater.* **17**, 3695-3704 (2005)).

Fig. 15 (a) C 1s and (b) F 1s XPS spectra of NCAI-LAO and NC91 after 100 cycles.

Fig. 16 Al 2p XPS spectra of NCAI-LAO after 100 cycles.

According to previous studies (*ACS Appl. Mater. Interfaces* **11**, 14095-14100 (2019); *Adv. Mater.* **24**, 1192-1196 (2012)), this fluorination can effectively scavenge HF in electrolyte and the resultant AlF_3 can promote the formation of stable cathode-electrolyte interface (CEI) films, which are beneficial for inhibiting dissolution of transition metals. In the revised manuscript, we have added the relevant discussion in the **Supplementary information** as follows:

*“Furthermore, the existence of Al-F signal in F 1s region (ca 686.5 eV in **Supplementary Fig. 15b**) and Al 2p region (ca 76.5 eV in **Supplementary Fig. 16**) of NCAI-LAO after cycling indicate that the LiAlO_2 layer can be partly fluorinated to form AlF_3 during electrochemical process¹⁰. A similar phenomenon was also observed when using another Al-containing coating^{11,12}. According to previous studies^{13,14}, this fluorination can effectively scavenge HF in electrolyte and the resultant AlF_3 can promote the formation of stable cathode-electrolyte interface (CEI) films, which are beneficial for inhibiting dissolution of transition metals.”* (Page 16 in **Supplementary information**)

4. Impedance data in Fig. 13 in Supp Info: EIS data should always be presented in squared graphs. Please, reshape the graphs.

Reply: We highly appreciate the valuable suggestions. In the revised manuscript, the graphs of EIS data have been reshaped as follows:

Fig. 17 (a, b) Nyquist plots after different cycles of NCAI-LAO and NC91, and the insets show the equivalent circuits for the impedance spectra. (c, d) Change of R_{sf} and R_{ct} for NCAI-LAO and NC91 with respect to the cycle number.

5. Authors should add some information in the experimental part such as:

- Value of $1 C$ considered.
- Amount of electrolyte used in coin cell and in the final pouch cell.
- Thickness and porosity of the electrodes employed in the final pouch cell.

Reply: We are sorry for missing this details. In the revision, we have added the detailed informations of experimental part as follows:

- “The value of $1 C$ was 180 mA g^{-1} ” (**Page 21, Line 14**)
- “Amounts of electrolyte used in coin cell and in pouch cell are $100 \mu\text{L}$ and 8 g , respectively”. (**Page 21, Line 11-12**)
- “Thickness and porosity of the electrodes employed in the final pouch cell are $113 \mu\text{m}$ and 42% respectively.” (**Page 21, Line 4-5**)

Overall, I particularly enjoyed reading this manuscript and I believe the results reported might be beneficial to a broad audience.

Reply: We would like to express our warm thanks regarding your recognition and positive recommendation for our work.

To Reviewer #3:

Comments: This manuscript demonstrates a gradient Al-doped and LiAlO₂-coated high-nickel cathode by using a new deposition method with oxalate ligands. The authors used calculations to show that Al³⁺ preferentially moves to the tetrahedral interstices compared with Ni²⁺ which eliminates cation mixing. Electrochemical performance of the coated cathode showed superb cyclability and improved power density. However, migration of Al³⁺ to the tetrahedral site has been suggested before for several times (Chem. Mater. 2003, 15, 4476-4483, J. Power Sources, 2003, 115, 305-314). As authors' suggestion even requires more obvious experimental data, I don't feel that this work is a substantial progress to be published in Nature Communications as compared to the above prior arts. The following points are encouraged to address:

Reply: We highly appreciate the comments from the reviewer, which improve the quality of this work during the revision.

The previous works suggested by the reviewer predicted that the potential migration of Al³⁺ to the tetrahedral site, and there is still a lack of theoretical and experimental evidences. Besides, it is still challenging but significant to establish a unequivocal correlation between the migration of Al³⁺ and electrochemical performance. More significantly, the novelty of this work is to effectively overcome the self-nucleation reactions of Al compounds through the assistance of oxalate, further to synchronously achieve LiAlO₂-coating layer on the surface of secondary particles and high-efficiency gradient Al-doping inside the primary particles. This design can address two key issues of crystal disintegration and interfacial instability to develop high-energy and power-dense Ni-rich cathodes with a long life and good thermal stability. Instead, the statement of Al³⁺ migration to the tetrahedral site is to explain the enhancement mechanism of electrochemical performance, despite of the absence of experimental evidences in the original manuscript. However, in the revision, double aberration-corrected STEM characterizations were utilized to identify that the occupy of Al³⁺ in tetrahedral site, as suggested by the reviewer. In addition, this occupancy behavior can effectively block Ni²⁺ migration to vacancies in the Li layer, as shown in the response to the first comments, which agrees well with our theoretical calculations. Furthermore, *in-situ* XRD and finite element analysis reveal the relationship between structure change and electrochemical performance (**Fig. 4a-d**). The preferential occupancy of Al³⁺ in tetrahedral site can alleviate the migration of Ni²⁺ to Li layer, which can further mitigate the contraction of lattice parameter and corresponding H2-H3 phase transition to improve structure stability and reduce internal mechanical stress. The certified occupy behavior and revealed relationship with electrochemical performance promote the understanding of the Ni-rich cathode modification. Based on the disucssion above, we deem that the insights indicated in this work would shed new lights in the design of highly excellent Ni-rich cathodes.

1. The authors showed that Al³⁺ prefers to migrate to tetrahedral interstices in Li layer by calculations. However, it is required to add more experimental evidence, such as TEM or STEM images with FFT pattern to visualize that Al³⁺ ions truly exist in the tetrahedral sites at the charged state and block Ni²⁺ migration to vacancies in the Li layer.

Reply: Thanks for the reviewer's constructive suggestions. As suggested by the reviewer, the double aberration-corrected scanning transmission electron microscopy (Cs-STEM) with corresponding energy dispersive spectroscopy (EDS) was employed to precisely characterise

the structure of the NCAI-LAO after being charged at 4.3 V. The representative Cs-STEM images acquired in high-angle annular dark-field (HAADF) mode along with the [100] orientation clearly demonstrate the well-maintained layered structure. Considering that the atom contrasts in the HAADF images depend on the atomic number, the layers with bright dots can be assigned the transition metal (TM) layers, and those with the dark dots can be assigned to the Li layers. As schematically shown by the model in **Supplementary Fig. 11a**, the tetrahedral sites should be located at the gap of TM layer and Li layer. Then, Al element EDS line analysis was carried out along the white arrows between the TM layer and Li layer in the **Fig. S1** below. The line scan profile exhibit obvious peaks corresponding to Al, which visualizes the presence of Al^{3+} in the tetrahedral sites at the charged state. Furthermore, the fast Fourier transform pattern for the HAADF image reveals the hexagonal phase with the R-3m space group rather than the cubic phase with the Fm-3m space group. According to the previous studies (*Nat Commun* **5**, 3529 (2014); *Nano Energy* **54**, 313-321 (2018);), the migration of Ni^{2+} to vacancies in the Li layer should result in the phase transformation from hexagonal phase to the cubic phase. Based on the above evidences, we conclude that the presence of Al^{3+} in the tetrahedral sites suppress the migration of Ni^{2+} to the vacancies in the Li layers.

Fig. S1 STEM-HAADF image with corresponding EDS line analysis of Al between TM layer and Li layer.

Fig. 11 (a) The crystal structure of Ni-rich cathode in the [100] orientation and the schematic diagram of EDS line scan. (b) The FFT pattern of NCAI-LAO obtained from HAADF image in **Fig 2e**.

In the revised manuscript, we have added these results and the relevant discussion as follows:
“To assess the effect of Al doping on the electrochemical reaction of the Ni-rich cathode, the double aberration-corrected scanning transmission electron microscopy (Cs-STEM) with

corresponding energy dispersive spectroscopy (EDS) was employed to precisely characterise the structure of the NCAI-LAO after being charged at 4.3 V. The representative Cs-STEM images acquired in high-angle annular dark-field (HAADF) mode along with the [100] orientation clearly demonstrate the well-maintained layered structure (Fig. 2e). Considering that the atom contrasts in the HAADF images depend on the atomic number, the layers with bright dots can be assigned the transition metal (TM) layers, and those with the dark dots can be assigned to the Li layers³⁴. As schematically shown by the model (Supplementary Fig. 11a), the tetrahedral sites should be located at the gap of TM layer and Li layer. Then, Al element EDS line analysis was carried out along the white arrows between the TM layer and Li layer in the Fig. 2e. The line scan profile exhibit obvious peaks corresponding to Al, which visualizes the presence of Al³⁺ in the tetrahedral sites at the charged state (Fig. 2f). Furthermore, the fast Fourier transform (FFT) pattern for the HAADF image reveals the hexagonal phase with the R-3m space group rather than the cubic phase with the Fm-3m space group. Therefore, the presence of Al³⁺ in the tetrahedral sites suppress the migration of Ni²⁺ to the vacancies in the Li layers during electrochemical process, which can further alleviate the phase transformation from hexagonal phase to the cubic phase.” (Page 11, Line 4-21)

Fig. 2 (a) Schematic illustration of synchronous gradient Al-doped and LiAlO₂-coated LiNi_{0.9}Co_{0.1}O₂ cathode. (b) TEM images and corresponding EDS mapping of NC91-Al(OH)₃ and NCAI-LAO. (c) High-resolution TEM images of surface for NCAI-LAO. (d) STEM-HAADF image of internal particles and corresponding EDS element line distribution of Ni and Al. (e, f) Cs-STEM-HAADF image with corresponding EDS line analysis of Al between TM layer and Li layer.

2. Oxalate-mediated co-precipitation method, one of the novelties in this manuscript, has been reported before on cathode synthesis (*J. Mater. Chem. A*, 2015, 3, 9427, *J. Mater. Chem. A*, 2015, 3, 21219) with a strength that the anion balances chelating level of metal ions. Though the authors showed the propriety of using oxalate ligand for Al coating by calculations, the novelty of the method can't help being discolored.

Reply: Thanks for the comments from the reviewer. In previous works, such as the literature suggested by the reviewer, oxalate was sometimes used as an alternative of ammonium to achieve the co-precipitation of multiple metal ions in the synthesis of the precursors of Li-rich cathodes. However, the highlight of this work is to simultaneously obtain uniform LiAlO₂-coating layer on the surface of secondary particles and high-efficiency gradient Al-doping inside the primary particles, toward improving the electrochemical performance and thermal stability of the Ni-rich cathodes. To realize above-mentioned modification strategy, it is pivotal to achieve controllable and uniform surface deposition of Al-containing compounds on the surface of the precursors of Ni-rich cathodes, but this is also very difficult since aluminum compounds are prone to self-nucleation reactions in the solution. Therefore, a suitable ligand is required to balance the Al³⁺ complexation rate and the subsequent uniform surface deposition. DFT calculations were then performed to guide the choice of an appropriate ligand. Based on the calculated results, the oxalate molecule as the ligand can bind with Al³⁺ moderately, and the Al-oxalate complex can adsorb on the precursor easily, which could contribute to the preferential nucleation and growth on the precursor surface during precipitation. These predictions were further confirmed by our experimental results. Based on the discussion above, we deem that the oxalate is firstly utilized to realize better Al-element modified Ni-rich cathode in this work.

In the revised manuscript, we have added the relevant discussion as follows:

“Considering that the structural deterioration starts from the surface region, it is significant to simultaneously achieve uniform LiAlO₂ coating and high-efficiency gradient Al-doping for Ni-rich cathodes by a simple and scalable approach. To this end, it is pivotal to achieve controllable and uniform surface deposition of Al-containing compounds on the surface of the precursors of Ni-rich cathodes, but this is also very difficult since aluminum compounds are prone to self-nucleation reactions in the solution. Therefore, a suitable ligand is required to balance the Al³⁺ complexation rate and the subsequent uniform surface deposition. DFT calculations were then performed to guide the choice of an appropriate ligand to prepare a Ni-rich oxide precursor, which can help prepare the ideal aluminium-element-modified LiNi_{0.9}Co_{0.1}O₂ cathode (NC91)” (Page 6, Line 10-16)

3. In the electrochemical performance part, the coated cathodes showed small surface film resistance and reduced polarization, while the authors gave no demonstration why the kinetic properties of the cathodes were improved. It is generally reasonable that the impedance of the cathodes increases when coated. Please explain why.

Reply: We are sorry for making the reviewer confused. The improved kinetics properties after modification is ascribed from the surface LiAlO₂ coating and bulk Al³⁺ doping. The coating of LiAlO₂ layer can mitigate parasitic reactions at the interface to reduce the thickness of the cathode-electrolyte interface (CEI) films (**Supplementary Fig. 15**), and improve Li⁺ conductivity at electrode/electrolyte interface benefitted from high Li⁺ conductivity of LiAlO₂

(up to $3 \times 10^{-5} \Omega^{-1} \text{ cm}^{-1}$). Therefore, the initial surface film resistance (R_{sf}) is reduced after modification (**Supplementary Fig. 17c**). Besides, the improved structure integrity by the surface LiAlO_2 coating and bulk Al^{3+} doping can alleviate the formation of new exposed surface during cycle process, and thus inhibit the formation of new CEI films (the insets of **Fig. 4b, c**). Thereby, compared to the obvious increase in the R_{sf} of pristine NC91 during cycling, the R_{sf} of NCAI-LAO is almost unchanged (**Supplementary Fig. 17c**). Furthermore, the doped Al^{3+} can reduce Li/Ni disorder and increase the lattice parameters (**Supplementary Fig. 4 and Supplementary Table 5**), which can significantly enhance Li^+ conductivity inside the lattice (as shown in **Supplementary Fig. 13**). Consequently, the increase of Li^+ conductivity at the interface and inside the lattice can eventually reduce electrochemical polarization. In the **Supplementary information** of revision, we have added the relevant discussion to make the reader more clear as follows:

“The coating of LiAlO_2 layer can mitigate parasitic reactions at the interface to reduce the thickness of the cathode-electrolyte interface (CEI) films and meanwhile improve Li^+ conductivity at electrode/electrolyte interface benefitted from high Li^+ conductivity of LiAlO_2 (up to $3 \times 10^{-5} \Omega^{-1} \text{ cm}^{-1}$). Therefore, R_{sf} is reduced after modification. Furthermore, the improved structure stability and particle integrity by the surface LiAlO_2 coating and bulk Al^{3+} doping can alleviate the formation of new exposed surface during cycle process, and thus inhibit the formation of new CEI films. Compared to the obvious increase in the R_{sf} of pristine NC91 during cycling, the R_{sf} of NCAI-LAO is almost unchanged.” (Page 17 in Supplementary information)

“The reduced electrochemical polarization and improved Li^+ diffusion coefficients are ascribed from LiAlO_2 coating and Al^{3+} doping. The doped Al^{3+} can reduce Li/Ni disorder and increase the lattice parameters, which can significantly boost Li^+ transfer inside the lattice. Besides, the coating of LiAlO_2 with high Li^+ conductivity is capable for improving Li^+ transfer at the interface.” (Page 13 in Supplementary information)

4. According to in-situ XRD data, coating and gradient doping helps to reduce H2-H3 transition and suppress microcracks. However, there is no clear correlation between cation mixing and H2-H3 transition. Besides, it is still faint how surface coating and doping can even control the bulk phenomena such as H2-H3 transition and microcrack formation. In addition, according to Fig. 2f and Supplementary Fig. 6, Al^{3+} rarely exists more than 60 nm away from the grain boundaries. In the region without Al^{3+} , it is wondering how Ni^{2+} cation mixing can be eliminated.

Reply: We are sorry for making the reviewer confused. Considering the cation mixing in Ni-rich cathodes, the oxidation of Ni^{2+} to Ni^{4+} in the Li layer at the end of charging process will cause prominent contraction of the lattice along the c-axis because of the short bond length of the Ni^{4+} -O bond, which results in the H2-H3 phase transition (*J. Power Sources* **68**, 120-125 (1997); *J. Electrochem. Soc.* **147**, 1314-1321 (2000)). In this work, the Al^{3+} doping can mitigate cation mixing to reduce the content of Ni^{2+} and Ni^{4+} in the Li layer, which can further assuage the contraction of c-axis and thus alleviate the H2-H3 transition during charging/discharging process, as revealed by the experimental results in **Fig. 4a-c**. In addition, the lattice contraction caused by the oxidation of Ni^{2+} will cause anisotropic volume change of the primary particles and subsequently generate internal mechanical stresses, which is considered as the origin of crack formation. Therefore, the coating and the gradient Al^{3+} doping can mitigate the H2-H3

transition to reduce the mechanical stress and crack formation (**Fig. 4d** and the insets inside **Fig. 4 b, c**).

Fig. 4 (a) Three-dimensional contour plots of (003) peaks from *in situ* XRD patterns for NC91 and NCAI-LAO with corresponding capacity vs. potential curves. Variations of (b) c-axis lattice parameters and (c) unit cell volume as a function of charging potential, and insets show cross-sectional SEM images of electrodes after 100 cycles. (d) Distributions of volume deformation, corresponding *Von mises* stress, tensile and compressive stresses through NCAI-LAO and NC91 when charging to 4.3 V. (e) Phase-transition temperature of NCAI-LAO and NC91 from HT-XRD patterns. (f) Differential scanning calorimetry profiles of NCAI-LAO and NC91 in delithiated state (charged to 4.3 V).

In addition, although Al^{3+} content showed a gradient distribution, around 2.0 at% of Al^{3+} is still existed in the center of primary particles, as shown in figure below. According to previous works (*Nat. Mater.* **20**, 84-92 (2021); *Adv. Energy Mater.* **10**, 2001035 (2020)), Ni^{2+} cation mixing and structure disintegration firstly start from the outer surface. For the interior of particles which are more stable than the surface, a small amount of Al^{3+} is capable for alleviating Ni^{2+} cation mixing.

Fig. S2 (a) STEM-HAADF image of internal particles and corresponding EDS element line distribution of Ni and Al. (b) Relative content of Al as a function of etching depth based on XPS of NCAI-LAO, inset in (b) shows STEM-EDS element mapping. (c) The Al^{3+} distribution across the secondary particles of NCAI-LAO detected by EDS point analyses. The white triangles (numbers are indicated) mark different point for EDS analysis in a single microsphere.

In the revision, we have added the relevant discussion to make the reader more clear as follows:

“Ni²⁺ in the Li layer caused by cation mixing was oxidized to Ni⁴⁺, generating NiO₂ sheets at a high charging state. These would shrink the Li layer like the TM layer and eventually cause c-axis contraction in the lattice, which is considered as a feature of H2-H3 phase transition^{33,34}”
(Page 15, Line 11-14)

“The contraction of lattice along the c-axis and the shrinkage of unit cell volume will cause anisotropic volume change of the primary particles to generate internal mechanical stresses, which is considered as the origin of crack formation.” **(Page 16, Line 9-11)**

“According to previous work^{17,18}, Ni²⁺ cation mixing and structure disintegration firstly start from the outer surface. For the interior of particles which are more stable than the surface, a small amount of Al³⁺ is capable for alleviating Ni²⁺ cation mixing.” **(Page 10, Line 16-19)**

REVIEWERS' COMMENTS

Reviewer #2 (Remarks to the Author):

Authors have carefully addresses all points raised in teh first revision. The quality of the paper has further increased and the responses are clearly given. Publication of the manuscript in its present form is suggested.

Reviewer #3 (Remarks to the Author):

After the first revision, the authors reinforced the novelty of the manuscript by acquiring experimental evidence and adding the explanation in detail. Especially, STEM-EDS line analysis of Al figure obviously plays an important role in the manuscript to show an experimental evidence that Al³⁺ ions exist at the tetrahedral site of the Li slab. Moreover, the authors supplemented the body to explain the correlation between cation mixing and H₂-H₃ transition, enabling XRD data and finite element analysis to show the merit of simultaneous Al coating and doping process. I support the manuscript to be published in Nature Communications, after addressing yet remained point below:

1. In Fig. 2f, only one Al³⁺ ion was detected in 5 nm of the layered structure. I support the authors' opinion that the Al³⁺ doping obviously assists to mitigate the cation mixing, however, I'm still wondering that how the region between two doped Al³⁺ ions also can maintain the layered structure with no cation mixing. Please mention the additional clarification in the body.

To Reviewer #1:

Comments: Authors have carefully addressed all points raised in the first revision. The quality of the paper has further increased and the responses are clearly given. Publication of the manuscript in its present form is suggested.

Reply: We appreciate the reviewer for the positive comments and the previous valuable suggestions.

To Reviewer #2:

Comments: After the first revision, the authors reinforced the novelty of the manuscript by acquiring experimental evidence and adding the explanation in detail. Especially, STEM-EDS line analysis of Al figure obviously plays an important role in the manuscript to show an experimental evidence that Al^{3+} ions exist at the tetrahedral site of the Li slab. Moreover, the authors supplemented the body to explain the correlation between cation mixing and H2-H3 transition, enabling XRD data and finite element analysis to show the merit of simultaneous Al coating and doping process. I support the manuscript to be published in Nature Communications, after addressing yet remained point below:

1. In Fig. 2f, only one Al^{3+} ion was detected in 5 nm of the layered structure. I support the authors' opinion that the Al^{3+} doping obviously assists to mitigate the cation mixing, however, I'm still wondering that how the region between two doped Al^{3+} ions also can maintain the layered structure with no cation mixing. Please mention the additional clarification in the body.

Reply: Thanks very much for the kind questions. The amount of doped Al^{3+} is estimated to around 2.0 at. % by the EDS analysis in Fig.2d, f. Notably, as revealed by *in-situ* XRD (Fig. 4a-c) and Cs-STEM (Fig. 2e) characterizations, the doped Al^{3+} ions can effectively suppress the cation mixing and thus alleviate the shrinkage of cell parameters and the degradation of the layered structure. It seems to be elusive that how the region between two doped Al^{3+} ions also can maintain the layered structure. Note that only parts of nickel ions rather than all of nickel ions in Ni-based cathode materials will transfer to Li vacancies after Li^+ extraction (*J. Mater. Chem.* **11**, 131-141 (2001), *Energy Environ. Sci.* **11**, 1271-1279 (2018)). We speculate that the part of nickel ions preferring to transfer to Li vacancies could be limited by the doped Al^{3+} ions, while the other part of nickel ions that not easily transfer to Li layer can stay at the Ni layer and thus maintain the layered structure. Therefore, such concentration of Al^{3+} ions could effectively assist to mitigate the cation mixing and stabilize the layered structure. To make the readers clearer, we have added the relevant discussion in the revised manuscript as follows:

“Note that only parts of nickel ions rather than all of nickel ions in Ni-based cathode materials will transfer to Li vacancies after Li^+ extraction, in which the part of nickel ions preferring to transfer to Li vacancies could be limited by the doped Al^{3+} ions, while the other part of nickel ions that not easily transfer to Li layer can stay at the Ni layer. Therefore, the region even between two doped Al^{3+} ions can also maintain the layered structure.” (Page 14, Line 2-7)